# Integrating Parental Phenotypic Data Enhances Prediction Accuracy of Hybrids in Wheat Traits

**DOI:** 10.3390/genes14020395

**Published:** 2023-02-02

**Authors:** Osval A. Montesinos-López, Alison R. Bentley, Carolina Saint Pierre, Leonardo Crespo-Herrera, Josafhat Salinas Ruiz, Patricia Edwigis Valladares-Celis, Abelardo Montesinos-López, José Crossa

**Affiliations:** 1Facultad de Telemática, Universidad de Colima, Colima 28040, Mexico; 2International Maize and Wheat Improvement Center (CIMMYT), Km 45, Mexico City 52640, Mexico; 3Colegio de Postgraduados Campus Córdoba, Km. 348 Carretera Federal Córdoba-Veracruz, Amatlán de los Reyes, Veracruz 94946, Mexico; 4Bachillerato 22, Universidad de Colima, Cuauhtémoc, Colima 28510, Mexico; 5Centro Universitario de Ciencias Exactas e Ingenierías (CUCEI), Universidad de Guadalajara, Guadalajara 44430, Mexico; 6Colegio de Postgraduados, Montecillos 56230, Mexico

**Keywords:** genomic prediction, parental information, prediction accuracy, correlated traits

## Abstract

Genomic selection (GS) is a methodology that is revolutionizing plant breeding because it can select candidate genotypes without phenotypic evaluation in the field. However, its practical implementation in hybrid prediction remains challenging since many factors affect its accuracy. The main objective of this study was to research the genomic prediction accuracy of wheat hybrids by adding covariates with the hybrid parental phenotypic information to the model. Four types of different models (MA, MB, MC, and MD) with one covariate (same trait to be predicted) (MA_C, MB_C, MC_C, and MD_C) or several covariates (of the same trait and other correlated traits) (MA_AC, MB_AC, MC_AC, and MD_AC) were studied. We found that the four models with parental information outperformed models without parental information in terms of mean square error by at least 14.1% (MA vs. MA_C), 5.5% (MB vs. MB_C), 51.4% (MC vs. MC_C), and 6.4% (MD vs. MD_C) when parental information of the same trait was used and by at least 13.7% (MA vs. MA_AC), 5.3% (MB vs. MB_AC), 55.1% (MC vs. MC_AC), and 6.0% (MD vs. MD_AC) when parental information of the same trait and other correlated traits were used. Our results also show a large gain in prediction accuracy when covariates were considered using the parental phenotypic information, as opposed to marker information. Finally, our results empirically demonstrate that a significant improvement in prediction accuracy was gained by adding parental phenotypic information as covariates; however, this is expensive since, in many breeding programs, the parental phenotypic information is unavailable.

## 1. Introduction

The question of how to feed the growing world population is not new, and there is evidence that several thousand years ago, farmers had already begun to genetically and physically modify plants to achieve better yields. In the 21st century, however, food security is not only threatened by a rapidly growing population, but also by climate change, which limits natural resources such as water, fuel, minerals, and arable land and contributes to the elevation of emissions with greenhouse gas potential. Plant breeding is considered by many experts as the starting point for the human food chain, as it significantly contributes to producing high and stable yields with low external inputs of non-renewable resources, low greenhouse-gas emissions, and a low concentration of undesirable substances [1].

Meuwissen [2] proposed improving the efficiency of plant-breeding through a predictive methodology called Genomic Selection (GS), which can select the best candidates without phenotypic information. This is possible because a reference population with phenotypic and genotypic information can be trained using a statistical machine-learning model that is subsequently used to make predictions for candidate lines (target population) that were only genotyped [3]. This methodology is transforming plant-breeding and is being implemented for many crops like wheat, maize, cassava, rice, chickpea, and groundnut, among others [4,5,6,7]

For GS to be successfully implemented, an accurate prediction model must be based on a reference population comprised of individuals with both genotypic and phenotypic data to predict unobserved cultivars with only genotypic information. Extensive research studies have been conducted and novel statistical methods that incorporate pedigree, genomic, and environmental covariates (e.g., weather data) into statistical–genetic prediction models have been developed [8]. Genomic Best Linear Unbiased Predictor (GBLUP) models are widely used in GS, and the extension of GBLUP for incorporating genotype × environment interaction (GE) has improved the accuracy of predicting unobserved cultivars in environments. Jarquin et al. [9] found that the prediction accuracy of models, including the GE term, was substantially higher (17–34%) than models based only on main effects. For a maize data set with an ordinal response variable, Montesinos-López et al. [10] reported that models that included GE achieved gains of 9–14% in prediction accuracy over models that only included main effects. Using wheat data, Cuevas et al. [11] observed that models with the GE term were up to 60–68% better in terms of GP accuracy than the corresponding single-environment models.

Prediction of hybrid performance is fundamentally important in modern hybrid breeding programs, and the best linear unbiased prediction (BLUP) model has been found useful for predicting the performance of unobserved single crosses using the performance of observed single crosses based on the pedigree (i.e., coancestry coefficient) relationship between the inbred lines forming the unobserved and observed single crosses. When studying and assessing hybrid performance, two sources of variation are important: the estimation of the additive effects among lines based on the variance of the general combining ability and the dominance and/or epistatic effects among the lines based on the variance of the specific combining ability of the cross between lines.

In the context of hybrid development, GS can provide significant savings in resources, since only the parental information is required; the genotypes of hybrids can be deduced from the genotypes of their parents rather than sequenced anew, which significantly reduces the cost of hybrid development [12]. However, it is challenging to identify the best hybrids, since many combinations of parental information should be evaluated: a process that significantly increases the cost. For this reason, to improve the efficiency of the GS methodology and take advantage of the information available during hybrid development, some authors have proposed the integration of parental information in genomic prediction models [13,14]. The approach for integrating the parental information in the genomic prediction models used by Liang et al. [13] and Jarquin et al. [15] was to increase the training set with the parental phenotypic information; that is, the training set was extended by including the phenotypic and genotypic data for the inbred lines. Conversely, the approach of Xu et al. [14] consisted of integrating the parental information in the prediction models as covariates computed from the parents of each hybrid while maintaining a fixed-size training set. The results of Liang et al. [13] were mixed: in some traits, there was a slight improvement in prediction accuracy, while in others, a decrease in prediction accuracy was observed. They concluded that “the naive incorporation of inbred genotype and trait information into training datasets decreased prediction accuracy for high heterosis traits” [13]. The results of Jarquin et al. [15] also showed no clear advantage in prediction performance by adding the parental phenotypic information into the training set. However, the results of Xu et al. [14] reported an increase in prediction accuracy of 13.6%, 54.5%, 19.9%, and 8.3%, for GY, number of tillers per plant, number of grains per panicle, and 1000 grain weight, respectively, showing a consistent gain in terms of prediction accuracy.

We propose using parental phenotypic information as covariates like Xu et al. [14] in genomic prediction models to improve the prediction accuracy of the genomic selection methodology. Four types (groups) of models were evaluated with these two types of parental information covariates using a wheat data set from the International Maize and Wheat Improvement Center (CIMMYT). These four types of models are MA, MB, MC, and MD. We further proposed two ways of adding the parental phenotypic information as covariates to the genomic prediction models: (a) using only the parental information of the trait to be predicted (MA_C, MB_C, MC_C, and MD_C) and (b) using both parental information of the trait to be predicted and the parental information other correlated traits (MA_AC, MB_AC, MC_AC, and MD_AC).

## 2. Materials and Methods

### 2.1. Phenotypic Data

A total of 1888 hybrids obtained by crossing 667 females and 18 males were evaluated in field experiments for three years at CIMMYT’s Campo Experimental Norman E. Borlaug (CENEB or the Norman E. Borlaug Experiment Station) near Ciudad Obregon, Sonora, Mexico. The number of hybrids evaluated during the winter growing seasons in 2014 to 2015 (Year 1), 2015 to 2016 (Year 2), and 2016 to 2017 (Year 3) were 703, 655, and 1197, respectively, with 225 and 383 common hybrids in each consecutive year. The elite female and male parents were chosen from CIMMYT’s spring bread wheat program based on their performance for the traits of interest, suitability for producing hybrids, and ancestral diversity as measured with a coefficient of parentage [16].

Using a chemical hybridizing agent provided by Syngenta Inc., the hybrids were produced in alternate male and female strip plots measuring 6.4 m. Parents and hybrids were evaluated in α-lattice trials with two replications in two years. To guarantee a uniform plant density, 1000 seeds were sown in 4.8 m yield trial plots. In a high-yield-potential environment, the trials were conducted with four supplementary irrigations using standard agronomic practices. In all trials, males and females were planted together with the hybrids and two checks. Days to flowering (DTF), days to heading (DTH), days to maturity (DTM), grain yield (GY), and plant height (PHT) per plot were recorded for each entry. Phenotypic data were analyzed by using a mixed linear model implemented in META-R software [17], where genomic best linear unbiased predictions (BLUEs) were estimated after fitting the model with trial, genotype, and replication nested within trials, and sub-blocks nested within trials and replications. BLUEs were obtained for each hybrid and parent and used for further analyses [16]. Three traits were analyzed in this paper: GY, DTF, and DTH.

### 2.2. Genotypic Data

The 18 male and 667 female parents were genotyped using the Illumina iSelect 90 K Infinitum SNP genotyping array in the first year and the Illumina Infinium 15 K wheat SNP array (TraitGenetics GmbH) in the second and third years. A total of 13,005 single-nucleotide polymorphisms (SNPs) remained after combining the three datasets. Markers with <15% missing values were kept; then, after cleaning, the remaining missing markers were imputed using the mean allele frequency of the wheat lines with that specific marker. After imputation, markers with <0.05 minor allele frequency were removed. A total of 10,250 markers were used for further analysis. Although a larger set of hybrids and parents were evaluated in the field, only hybrids derived from SNP-genotyped parents were used for genomic predictions and different numbers of hybrids were included in each year [16].

### 2.3. Statistical Model

In this study, we evaluated four different types of models: Type A, B, C, and D (MA, MB, MC, and MD), where Types A and C did not include genomics, and Types B and D did. Models could include either one covariate (same trait to be predicted) (MA_C, MB_C, MC_C, and MD_C) or several covariates (of the same trait and other correlated traits) (MA_AC, MB_AC, MC_AC, and MD_AC). As shown below, while Type A and B models are similar, (A) is without genomics, and (B) is with genomics. Type C and D models are also similar, but (C) is without genomics, and (D) is with genomics.

#### 2.3.1. Model MB_AC

This model is given by
(1)$$Y=Z_{E}\beta_{E}+Z_{M}g_{M}+Z_{F}g_{F}+Z_{H}h+u_{M}+u_{F}+u_{H}+X_{AC}\beta_{AC}+\epsilon$$
where Y is the response vector (i.e., the hybrids’ adjusted phenotypic information); $Z_{E}$ is the design matrix for environments (year); $\beta_{E}$ is the vector of environmental effects: $\beta_{E}$ $∼N_{I}\left(0,\sigma_{E}^{2}I\right)$; $g_{M}$ is the vector of random effects due to the general combining ability (GCA) of markers for paternal lines (males, M); $g_{F}$ is the vector of random effects due to the GCA of markers for maternal lines (females, F); and h is the vector of SCA random effects for the crosses (hybrids, H). The incidence matrices $Z_{M}$, $Z_{F}$, and $Z_{H}$ relate Y to $g_{M}$, $g_{M}$, and h with $g_{M}∼N\left(0,\sigma_{M}^{2}G_{M}\right)$, $g_{F}∼N\left(0,\sigma_{F}^{2}G_{F}\right)$, and $h∼N\left(0,\sigma_{h}^{2}H\right)$, where $\sigma_{M}^{2}$, $\sigma_{F}^{2}$, and $\sigma_{H}^{2}$ are variance components associated with GCA and SGA; $G_{M}$, $G_{F}$, and H are relationship matrices for parental and maternal lines and hybrids, respectively. Finally, $\epsilon ∼N\left(0,\sigma_{\epsilon }^{2}I\right)$, where $\sigma_{\epsilon }^{2}$ is the variance associated with the residuals. The relationship matrices $G_{M}$ and $G_{F}$ were computed using markers [18]. Let $X_{m}$, m∈ {Male, Female} be the matrix of markers, and let $W_{m}$, be the matrix of centered and standardized markers. Then, $G_{m}=\frac{W_{m}W_{m}^{T}}{p}$ [16,19,20], where p is the number of markers and $H=G_{M}⊗G_{F}$, where ⊗ denotes the Kronecker product. While $u_{M}∼N\left(0,\sigma_{ME}^{2}V_{M}\right)$, $u_{F}∼N\left(0,\sigma_{FE}^{2}V_{F}\right)$, $u_{H}∼N\left(0,\sigma_{HE}^{2}V_{H}\right)$; $\sigma_{ME}^{2}$, $\sigma_{FE}^{2}$, and $\sigma_{hE}^{2}$ are variance components associated with male × environment, female × environment, and hybrid × environment interactions, respectively; and $V_{M}$, $V_{F}$, and $V_{H}$ are the associated variance–covariance matrices. The variance–covariance matrix is given by $V_{M}=Z_{M}G_{M}Z_{M}^{T}\#Z_{E}Z_{E}^{T}$, $V_{F}=Z_{F}G_{F}Z_{F}^{T}\#Z_{E}Z_{E}^{T}$, and $V_{H}=Z_{H}HZ_{H}^{T}\#Z_{E}Z_{E}^{T}$, where # stands for the Hadamard product. $X_{AC}$ is the matrix that contains the parental covariates of the trait to be predicted and of correlated traits. We computed two covariates from each trait using the parental phenotypic information. One covariate that captures the additive part is computed as
$$X_{AC,t,a}=\frac{\left(P_{M,t}+P_{F,t}\right)}{2}$$
where t=GY,DTFandDTH,anda denotes additive, $P_{M,t}$ is the phenotypic value of the parental male line for the *t*^th^ trait, and $P_{F,t}$ is the phenotypic value of the parental female line for the *t*th trait where the male and female are assumed to belong to different heterotic groups. The other covariates capture the dominance part, and it is computed as the absolute value of $P_{M,t}+P_{F,t}$($\left|P_{M,t}+P_{F,t}\right|$) for the *t*th trait:$$X_{AC,t,d}=\frac{\left|P_{M,t}+P_{F,t}\right|}{2}$$where d denotes dominance. The matrix $X_{AC}$ contains 6 columns, since two covariates (one for a and the other for d) were computed for each of the three traits under study. It is also important to point out that when the matrix of covariates $X_{AC}$ was ignored in the predictor, the model is denoted as **MB**, but when the matrix $X_{AC}$ only contained the information of one trait that corresponded to the trait to be predicted, this model is named **MB_C**, and the covariates are denoted as $X_{C}$. To facilitate the understanding and comparison, all these models described (**MB**, **MB_C,** and **MB_AC**) will be called Type B models.

#### 2.3.2. Model MA_AC

The predictor of model **MA_AC** is exactly equal to the predictor of model **MB_AC** (equation 1), but without markers information because (a) the vector of **random** effects due to the GCA of markers for paternal lines; (b) the vector of random effects due to the GCA of markers for maternal lines; and (c) the vector of SCA random effects for the hybrids (crosses) are distributed as $g_{M}∼N\left(0,\sigma_{M}^{2}I_{M}\right)$, $g_{F}∼N\left(0,\sigma_{F}^{2}I_{F}\right)$, and $h∼N\left(0,\sigma_{H}^{2}I_{H}\right)$ respectively. The distribution of interaction terms are $u_{M}∼N\left(0,\sigma_{ME}^{2}Z_{M}Z_{M}^{T}\#Z_{E}Z_{E}^{T}\right)$, $u_{F}∼N\left(0,\sigma_{FE}^{2}Z_{F}Z_{F}^{T}\#Z_{E}Z_{E}^{T}\right)$, $u_{H}∼N\left(0,\sigma_{HE}^{2}Z_{H}HZ_{H}^{T}\#Z_{E}Z_{E}^{T}\right)$. Under model **MA_AC,** when the matrix of covariates $X_{AC}$ was ignored in the predictor, this model is denoted as **MA;** when the matrix $X_{AC}$ only contained the information of one trait, which corresponds to the trait to be predicted, it is named model **MA_C**, and the covariates are denoted as $X_{C}$. To facilitate the understanding and comparison of the models just described (**MA**, **MA_C,** and **MA_AC**), they will be called Type A models. It is important to point out that those models that end with **_C** use parental phenotypic information of the trait to be predicted as covariates, while those models that end with **_AC** in addition to the covariates of the parental phenotypic information of the same trait also use parental phenotypic information of other correlated traits.

#### 2.3.3. Model MD_AC

This model is given by
(2)$$Y=Z_{E}\beta_{E}+Z_{H}h+u_{H}+X_{AC}\beta_{AC}+\epsilon$$

Model **MD_AC** is equal to model **MB_AC** but without the main effects of paternal effects (males), maternal effects (females), and its interactions with environments. Under model **MD_AC,** when the matrix of covariates $X_{AC}$ was ignored in the predictor, the model is denoted as **MD**, but when the matrix $X_{AC}$ only contained the information of one trait that corresponded to the trait to be predicted, it is termed **MD_C,** where the covariates were denoted as $X_{C}$. Again, the models just described (**MD**, **MD_C,** and **MD_AC**) will be called Type D models.

#### 2.3.4. Model MC_AC

The predictor of model **MC_AC** is exactly equal to the predictor of model **MD_AC** (Equation 2), but without markers information. For this reason, the vector of SCA random effects for the hybrids (crosses) were distributed as $h∼N\left(0,\sigma_{h}^{2}I_{H}\right)$, and the distribution of the interaction term was $u_{H}∼N\left(0,\sigma_{HE}^{2}Z_{H}HZ_{H}^{T}\#Z_{E}Z_{E}^{T}\right)$. Under model **MC_AC,** when the matrix of covariates $X_{AC}$ was ignored in the predictor, we call this **Model MC**, but when the matrix $X_{AC}$ only contained the information of one trait that corresponded to the trait to be predicted, we call this **model MC_C,** where the covariates are denoted as $X_{C}$. By the same logic, these models (**MC**, **MC_C,** and **MC_AC**) will be called Type C models.

The implementation of these models was carried out in the R statistical software using the BGLR library [21].

### 2.4. Evaluation of Prediction Performance

In each of the twelve models, we implemented a type of cross-validation that mimicked real breeding strategies, called untested lines, in tested environments under seven-fold cross-validation [22]. For this reason, 7−1 folds were assigned to the training set and the remaining to the testing set until each of the 7 folds were used at least once in the testing set. Next, the average of the seven folds was reported as prediction performance using the mean square error (MSE) as a metric. To compare the prediction accuracies between models of the same type (Type A, Type B, Type C, and Type D models), the relative efficiencies in terms of MSE were computed as
$$RE_{MSE}=\frac{MSE_{MX}}{MSE_{MX\_Z}}$$
where $MSE_{MX}$ and $MSE_{MX\_Z}$ designate the MSE of model, where X=A,B,CandD and Z=CandAC, respectively. For other scenarios, the relative efficiency was computed as
$$RE_{MSE}=\frac{MSE_{MX\_C}}{MSE_{MX\_AC}}$$

Under both, if $RE_{MSE}>1,$ the best prediction performance in terms of MSE was obtained using method MX_Z (or MX_AC), but when $RE_{MSE}<1,$ the best method was $MX\left(orMX\_C\right)$. When $RE_{NRMSE}=1,$ both methods were equally efficient.

## 3. Results

The results are provided in the three main sections, an Appendix B, and Appendix A. Section 3.1 and Section 3.2 show the results of Type A and B models, while Section 3.3 provides a comparison between the 12 models (MA, MA_C, MA_AC, MB, MB_C, MB_AC, MC, MC_C, MC_AC, MD, MD_C, and MD_AC) implemented across years for three traits. To make the results interpretable and easy to read, we have shown the results from Type C and D models in the Appendix B. Note that all results reported are, under cross-validation, called untested lines in tested environments. Furthermore, five Appendix A are given in Appendix A to complement the results displayed in the five figures (Figure 1, Figure 2 and Figure 3 in the main text and Figure A1 and Figure A2 in the Appendix B). Appendix A show the predictions for each trait in each environment and across environments for each trait (global) for Type A and B models and across years in terms of mean squared error under untested lines in tested environments cross-validation strategies. Appendix A show the prediction performance for each trait in each environment and across environments for each trait (global) for Type C and D models in terms of mean squared error (MSE) under untested lines in tested environments across validation strategies. In each of the five Appendix A, the header identifies both the model (Type A, B, C, D) and the inclusion (or not) of one or more covariates.

### 3.1. Type A Models

For the Type A model, we can observe in Figure 1A that the REs from comparing model MA with MA_C for GY in terms of MSE for each year and across years were 1.135 (year 1), 0.962 (year 2), 1.179 (year 3), and 1.141 (Global). This indicates that the MA_C model outperformed the MA model in most of the years: 13.5% (year 1), 17.9% (year 3), and 14.1% (Global). The REs of comparing the MA with MA_AC for GY in terms of MSE for each year and across years were 1.136 (year 1), 0.969 (year 2), 1.162 (year 3), and 1.137 (Global) (Figure 1A). This indicates that MA_AC outperformed the MA model in all years, except year two: 13.6% (year 1), 16.2% (year 3), and 13.7% (Global). Finally, the REs from comparing the MA_C with MA_AC for GY in terms of MSE for each year and across years were 1.001 (year 1), 1.007 (year 2), 0.986 (year 3), and 0.996 (Global). This indicates that MA_AC slightly outperformed MA_C by 0.1% (year 1) and 0.7% (year 2). For more details, see Appendix A.

In Figure 1B, the RE is given in terms of MSE and compares model MA with model MA_C for trait DTF for each year and across years. As shown, the REs are 1.019 (year 1), 1.126 (year 2), 1.208 (year 3), and 1.169 (Global), indicating that model MA_C outperformed the MA model in all years: 1.9% (year 1), 12.6% (year 2), 20.8% (year 3), and 16.9% (Global). The REs from comparing model MA with MA_AC for trait DTF in terms of MSE for each year and across years were 1.036 (year 1), 1.136 (year 2), 1.197 (year 3), and 1.167 (Global). This indicates that MA_AC outperformed the MA model in all years: 3.6% (year 1), 13.6% (year 2), 19.7% (year 3), and 16.7% (Global). Finally, the REs from comparing model MA_C with MA_AC for DTF in terms of MSE for each year and across years were 1.016 (year 1), 1.008 (year 2), 0.991 (year 3), and 0.998 (Global). This indicates that the MA_AC model is quite like the MA_C model, since it was only better in two years: 1.6% (year 1) and 0.8% (year 2). For more details, see Appendix A.

We can observe in Figure 1C the RE from comparing model MA with MA_C for trait DTH in terms of MSE for each year and across years: 1.026 (year 1), 1.141 (year 2), 1.208 (year 3), and 1.176 (Global). This indicates that model MA_C outperformed the MA model in all years: 2.6% (year 1), 14.1% (year 2), 20.8% (year 3), and 17.6% (Global). The REs of comparing model MA with MA_AC for trait DTH in terms of MSE for each year and across years were 1.176 (year 1), 1.044 (year 2), 1.207 (year 3), and 1.178 (Global). This indicates that model MA_AC outperformed MA model in all years: 17.6% (year 1), 4.4% (year 2), 20.7% (year 3), and 17.8% (Global). Finally, the REs of comparing the MA_C with MA_AC for trait DTH in terms of MSE for each year and across years were 1.018 (year 1), 1.009 (year 2), 0.999 (year 3), and 1.002 (Global). This indicates that MA_AC outperformed the MA_C model by only a small margin: 1.8% (year 1), 0.9% (year 2), and 0.2% (Global). For more details, see Appendix A.

### 3.2. Type B Models

For the Type B models, we can observe in Figure 2A that the REs from comparing model MB with MB_C for GY in terms of MSE for each year and across years were 1.050 (year 1), 0.958 (year 2), 1.114 (year 3), and 1.055 (Global). This indicates that model MB_C outperformed model MB in all years except year 2, by 5.0% (year 1), 11.4% (year 3), and 5.5% (Global). The REs from comparing the MB with MB_AC for GY in terms of MSE for each year and across years were 1.052 (year 1), 0.973 (year 2), 1.101 (year 3), and 1.053 (Global) (Figure 2A). This indicates that the MB_AC model outperformed the MB model in all but the second year by 5.2% (year 1), 10.1% (year 3), and 5.3% (Global). Finally, the Res from comparing the MB_C with MB_AC for GY in terms of MSE for each year and across years were 1.002 (year 1), 1.016 (year 2), 0.988 (year 3), and 0.998 (Global). This indicates that MB_AC outperformed the MB_C model by only a small margin: by 0.2% (year 1) and 1.6% (year 2). For more details, see Appendix A.

Figure 2B displays the REs in terms of MSE from comparing model MB with MB_C for trait DTF for each year and across years: 1.002 (year 1), 1.084 (year 2), 1.180 (year 3), and 1.119 (Global). This indicates that model MB_C outperformed model MB in all years, by 0.2% (year 1), 8.4% (year 2), 18.0% (year 3), and 11.9% (Global). The REs from comparing the MB model with the MB_AC model for trait DTF in terms of MSE for each year and across years were 1.042 (year 1), 1.092 (year 2), 1.180 (year 3), and 1.129 (Global). This indicates that the MB_AC model outperformed the MB model in all years by 4.2% (year 1), 9.2% (year 2), 18.0% (year 3), and 12.9% (Global). Finally, the REs from comparing the MB_C with MB_AC for trait DTF in terms of MSE for each year and across years were 1.041 (year 1), 1.008 (year 2), 1.00 (year 3), and 1.009 (Global). This indicates that the MB_AC model outperformed MB_C only by 4.1% (year 1), 0.8% (year 2), and 0.9% (Global). For more details, see Appendix A.

Figure 2C shows that the REs from comparing model MB with MB_C for trait DTH in terms of MSE for each year and across years were 1.000 (year 1), 1.089 (year 2), 1.181 (year 3), and 1.120 (Global). This indicates that the MB_C model outperformed the MB model in all years except the first year by 8.9% (year 2), 18.1% (year 3), and 12.0% (Global). The REs from comparing the MB with MB_AC for trait DTH in terms of MSE for each year and across years were 1.047 (year 1), 1.088 (year 2), 1.186 (year 3), and 1.133 (Global). This indicates that the MB_AC model outperformed the MB model in all years by 4.7% (year 1), 8.8% (year 2), 18.6% (year 3), and 13.3% (Global). Finally, the REs from comparing the MB_C with MB_AC for trait DTH in terms of MSE for each year and across years were 1.047 (year 1), 0.999 (year 2), 1.004 (year 3), and 1.011 (Global). This indicates that the MB_AC model outperformed MB_C in most of the years: 4.7% (year 1), 0.4% (year 3), and 1.1% (Global). For more details, see Appendix A.

### 3.3. Comparison across Years

Figure 3A–C show the differences between the twelve models in terms of prediction performance using the MSE as metrics for the three traits evaluated. Figure 3D displays the RE when comparing models without parental phenotypic information versus models with parental phenotypic information of the same type in terms of MSE for GY trait across years, which were 1.141 (MA vs. MA_C), 1.055 (MB vs. MB_C), 1.514 (MC vs. MC_C), and 1.064 (MD vs. MD_C). This indicates that models with parental phenotypic information of the same type of outperformed models without parental information across years (Global) by 14.1% (MA vs. MA_C), 5.5% (MB vs. MB_C), 51.4% (MC vs. MC_C), and 6.4% (MD vs. MD_C). The RE of comparing models without parental phenotypic information versus parental information of the same type plus additional correlated traits in terms of MSE for GY trait across years were 1.137 (MA vs. MA_AC), 1.053 (MB vs. MB_AC), 1.551 (MC vs. MC_AC), and 1.060 (MD vs. MD_AC); that is, models with parental phenotypic covariates of the same type plus additional covariates of correlated traits outperformed models without covariates of parental phenotypic information across years (Global) by 13.7% (MA vs. MA_C), 5.3% (MB vs. MB_C), 55.1% (MC vs. MC_C), and 6.0% (MD vs. MD_C). For more details, see Appendix A.

In Figure 3E, we can observe that the REs from comparing models without parental phenotypic information with models with parental information of the same type in terms of MSE for trait DTF across years were 1.169 (MA vs. MA_C), 1.119 (MB vs. MB_C), 1.964 (MC vs. MC_C), and 1.106 (MD vs. MD_C); that is, models with parental phenotypic information outperformed models without parental information across years (Global) in all cases: 16.9% (MA vs. MA_C), 11.9% (MB vs. MB_C), 96.4% (MC vs. MC_C), and 10.6% (MD vs. MD_C). The REs from comparing models without parental phenotypic information with models with parental phenotypic information of the same type plus additional correlated traits in terms of MSE for trait DTF across years were 1.167 (MA vs. MA_AC), 1.129 (MB vs. MB_AC), 1.996 (MC vs. MC_AC), and 1.113 (MD vs. MD_AC). This indicates that models with parental phenotypic information of the same trait and additional correlated traits outperformed models without parental information across years (Global) by 16.7% (MA vs. MA_C), 12.9% (MB vs. MB_C), 99.6% (MC vs. MC_C), and 11.3% (MD vs. MD_C). For more details, see Appendix A.

Figure 3F shows that the REs from comparing models without parental information and with parental information of the same trait in terms of MSE for trait DTH across years were 1.176 (MA vs. MA_C), 1.120 (MB vs. MB_C), 1.955 (MC vs. MC_C), and 1.112 (MD vs. MD_C). This means that models with parental phenotypic information of the same trait outperformed models without parental phenotypic information across years (Global) by 17.6% (MA vs. MA_C), 12.0% (MB vs. MB_C), 95.5% (MC vs. MC_C), and 11.2% (MD vs. MD_C). Meanwhile, the REs from comparing models without parental phenotypic information with models with parental phenotypic information of the same type plus additional correlated traits in terms of MSE for trait DTH across years were 1.178 (MA vs. MA_AC), 1.133 (MB vs. MB_AC), 2.005 (MC vs. MC_AC), and 1.120 (MD vs. MD_AC). This indicates that models with parental phenotypic information of the same type and additional correlated traits outperformed models without parental phenotypic information across years (Global) by 17.8% (MA vs. MA_C), 13.3% (MB vs. MB_C), 105.0% (MC vs. MC_C), and 12.0% (MD vs. MD_C). For more details, see Appendix A.

## 4. Discussion

Genomic selection is a very attractive predictive methodology because candidate lines can be selected without phenotypic evaluation. However, its practical implementation is still challenging, because when all factors that affect its accuracy are not efficiently optimized, its results may contain a high degree of uncertainty. For this reason, it is necessary to continue research on this topic.

For this paper, we studied the incorporation of parental phenotypic information as covariates in genomic prediction models. Two approaches for incorporating the parental phenotypic information were studied. The first approach consisted of adding the parental information of the trait to be predicted only as a covariate, while the second used both the parental information of the trait and the parental phenotypic information of other correlated traits as a covariate. To capture the additive and dominance information of the parental information for each trait, two covariates were computed; (Pheno_male + Pheno_female)/2 and the absolute value of (Pheno_male + Pheno_female)/2 captures the additive and dominance part, respectively. Using these covariates, we evaluated four types of models that contain different main and interaction effects in the predictor, and observed the trait GY across years, models, and type of covariates. The addition of parental information improved the prediction accuracy in terms of mean square error by 19.68%, while in trait DHT, this improvement was 34.98% and in trait DFT, the improvement was 34.53%. However, it is important to note that the Type C models (MC, MC_C, and MC_AC) displayed the largest improvement in prediction accuracy when the parental phenotypic information was incorporated. For example, when the parental information was added only for the trait to be predicted, the gain in prediction accuracy was 81.1%, while when in addition to the trait to be predicted the parental phenotypic information of other correlated traits was incorporated, the gain in prediction performance in terms of mean square error increased to 85.06%.

Our results coincide with those reported by Xu et al. [14] when working with rice. The authors reported that incorporating parental phenotypic information into conventional genomic prediction models increased prediction accuracy by 13.6%, 54.5%, 19.9%, and 8.3% for GY, number of tillers per plant, number of grains per panicle, and 1000 grain weight, respectively. However, these differ greatly from the findings of Liang et al. [13], who obtained mixed results. In some traits, improvements in prediction accuracy were obtained, while in others, there was a decrease. In Jarquin’s study [15], only a modest increase in prediction accuracy was observed. Some of the difference in our results versus those of Liang et al. [13] and Jarquin et al. [15] are because of the integration of the parental phenotypic information. In our case, like Xu et al. [14], the parental information was incorporated as covariates in the prediction models, whereas with Liang et al. [13] and Jarquin et al. [15], the training set with the parental phenotypic information was enlarged. Another reason for different results can be attributed to our use of the mean square error as a metric for evaluating the prediction accuracy, while the studies of Liang et al. [13], Jarquin et al. [15], and Xu et al. [14] used the Pearson’s correlation.

Hybrid breeding is an efficient system to break the yield barriers in many crops. CIMMYT elite lines from the spring bread wheat breeding program are crossed and the resulting F1 hybrids are primarily evaluated in Mexico. However, as CIMMYT lacks distinct male and female breeding pools, the main challenge remains in the parental selection for hybrid production. Although the per se performance of inbreds, plant phenology, and cross-pollinating traits are the initial criteria to select parents for hybrid production and testing, there is a limitation on the number of combinations that we can produce and test in the field. Since the beginning of its hybrid wheat program, CIMMYT has been routinely using the coefficient of parentage (COP) as one of the parental selection criteria to maximize genetic diversity. The wheat hybrid study of Basnet [16] demonstrated the potential application of pedigree information and molecular marker data in predicting single-cross hybrid performance. Applying hybrid prediction, we envision selecting potential hybrid parents from a broader genetic germplasm pool and earlier in the breeding cycle after preliminary and elite yield trail evaluation. However, Basnet’s [16] study did not include any covariate using parental information to enhance hybrid prediction; it shows the increase in genomic prediction accuracy by using existing parental phenotypic data. Additional studies are required to analyze environmental studies together with parental phenotypic information.

Furthermore, results of this study show that adding the parental phenotypic information to the genomic prediction model enhances the prediction accuracy of the GS methodology. It is noted that the way that the parental phenotypic information is incorporated in the genomic prediction models is key to improving the prediction performance. We also found that only a small gain in prediction performance is reached when, in addition to the parental phenotypic information of the same trait to be predicted, we add the parental phenotypic information of other correlated traits. This result could be interpreted such that adding other correlated traits from parents as covariates can be helpful in increasing the prediction accuracy only in those cases where these other traits are highly correlated with the trait of interest (to be predicted); otherwise, if the degree of correlation is low, no further improvement on genomic prediction accuracy can be achieved.

Our research contributes to the growing empirical evidence that adding metabolic/biochemical traits, also called endophenotypes, as covariates can help increase the prediction accuracy of genomic prediction models, since these covariates complement the explanatory power of genetic markers when used as covariates in prediction models, as noted by Melandri et al. [23]. For this reason, Westhues [24] points out that “complementing genomic data with other “omics” predictors can increase the probability of success for predicting the best hybrid combinations using complex agronomic traits.”

## 5. Conclusions

We proposed the integration of parental phenotypic information as covariates in conventional genomic prediction models. We found empirical evidence that, when parental phenotypic information is considered as a covariate in genomic prediction models, the prediction accuracy improves. Across traits and types of parental covariate information, we observed a gain in terms of mean square error of 16.1% (average of models MA_C and MA_AC vs. model MA), 10.2% (average of models MB_C and MB_AC vs. model MB), 83.1% (average of models MC_C and MC_AC vs. model MC), and 9.9% (average of models MD_C and MD_AC vs. model MD) by adding the parental phenotypic covariate information. However, we did not observe significant differences between the two approaches by adding the parental phenotypic covariate information (only of the trait to be predicted and with the trait to be predicted and correlated traits). Even though these results are not conclusive, they provide empirical evidence that a significant gain in prediction accuracy can be obtained by adding parental phenotypic information as covariates, and as such, we encourage more research with other data sets to corroborate our empirical findings.

## Figures and Tables

**Figure 1 genes-14-00395-f001:**
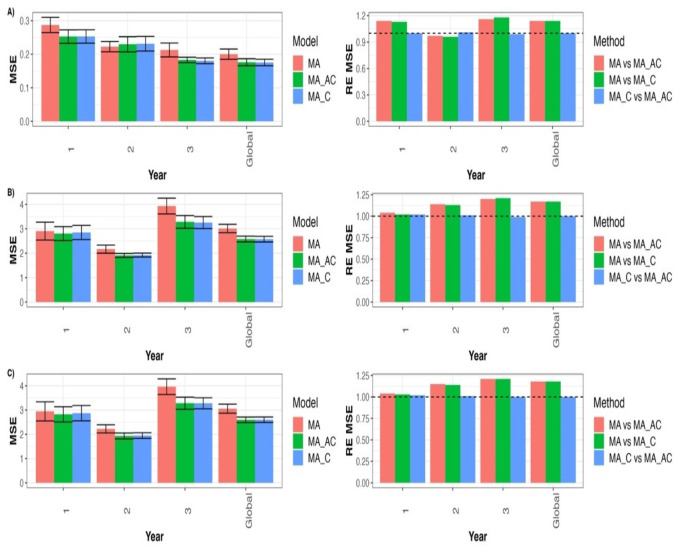
Prediction performance in terms of mean square error (MSE) for models MA, MA_C, and MA_AC for each trait in each environment and across environments (Global) under untested lines in tested environments cross-validation strategy. The relative efficiencies (RE_MSE) were computed to compare models MA_C vs. MA_AC, MA vs. MA_C, and MA vs. MA_AC. (**A**) for GY trait, (**B**) for DTF trait, and (**C**) for DTH trait.

**Figure 2 genes-14-00395-f002:**
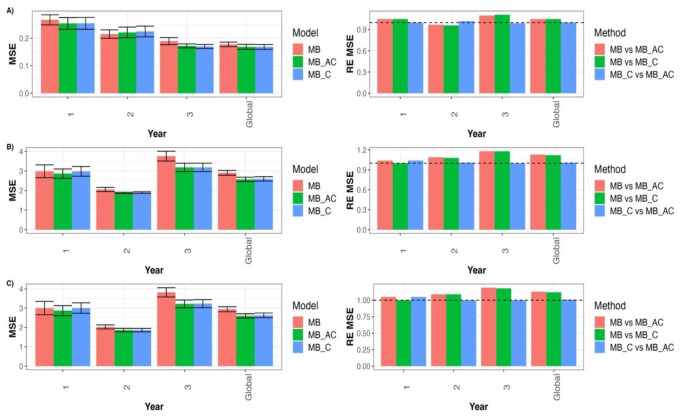
Prediction performance in terms of mean square error (MSE) for models MB, MB_C, and MB_AC for each trait in each environment and across environments (Global) under untested lines in a tested environment cross-validation strategy. The relative efficiencies (RE_MSE) were computed to compare models MB_C vs. MB_AC, MB vs. MB_C, and MB vs. MB_AC. (**A**) for GY trait, (**B**) for DTF trait, and (**C**) for DTH trait.

**Figure 3 genes-14-00395-f003:**
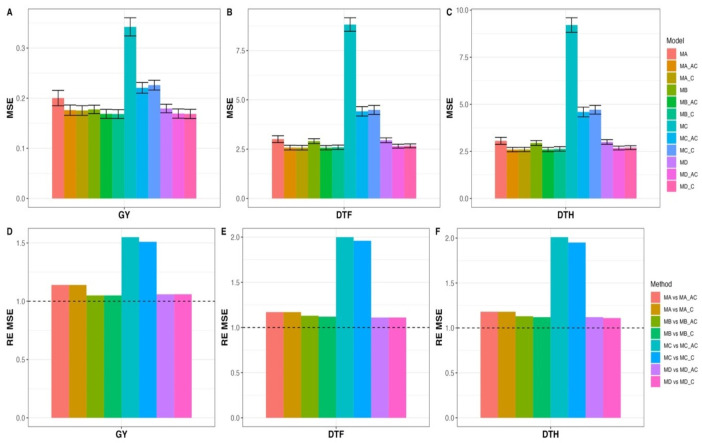
Prediction performance across environments (years) for each trait for all models—MA, MA_C, MA_AC, MB, MB_C, MB_AC, MC, MC_C, MC_AC, MD, MD_C, and MD_AC—in terms of mean squared error (MSE) under untested lines in a tested environment cross validation strategy. The relative efficiencies (RE_MSE) were computed by dividing the MSE of model MX by the MSE of model MX_C or MX_AC model, with X taking the values of (**A**–**D**). (**A**–**C**) correspond to traits GY, DTF, and DTH fitted to the 12 models shown on the right-hand side. (**D**–**F**) correspond to traits GY, DTF, and DTH comparing fitted to models shown on the right hand side.

## Data Availability

Phenotypic and genomic data can be downloaded from the link: http://hdl.handle.net/11529/10548129.

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
