# Peer review of "Integrating Parental Phenotypic Data Enhances Prediction Accuracy of Hybrids in Wheat Traits"

_genes, 2023, doi:10.3390/genes14020395_

Round 1

Reviewer 1 Report

under a dominance model absolute value of (Pheno_male+Pheno_female)/2.

Please confirm whether it is correct.

Author Response

RESPONSE TO REVISION ARTICLE GENES_2132017

RESPONSE to REVIEWER 1

Comments and Suggestions for Authors

under a dominance model absolute value of (Pheno_male+Pheno_female)/2.

Please confirm whether it is correct.

RESPONSEE: Many thanks for revising the article.  Yes, this is correct.

Reviewer 2 Report

Dear Authors,

The manuscript entitled "Integrating parental phenotypic data enhances prediction accuracy of hybrids in wheat traits" was revised. In general, the study can be useful for researchers working in the related field; however, in the material and methods section, only the results of the methods applied for "phenotypic" and "genotypic" data are given. The mention of these methods is necessary, or at least references to these applications should be cited. For example, SNP is mentioned, but only results are in the text. Why was the SNP chosen? Which primers were used? How was the protocol implemented? A publication on genes should answer these questions. Also, for some readers of the "genes" journal, this section is explained in a very complex way, and it should be given in a more understandable way. The study seems to address the field of "data mining," where statistical data are evaluated rather than genetic approaches. It would be more appropriate to publish the manuscript after the rearrangement mentioned in the report.

 Kind regards

This revisions described below;

 Title

OK

Abstract

Line 5-6, …..and additive model of (Pheno_male+Pheno_female)/2…

Line 10, it's been repeated here …..

….at least 14.1%, 5.5%, 51.4%, and 6.4% of (MD vs MD_C)…. seems more appropriate…

# Similarly, this arrangement can be made in the text.

Keywords

OK

Introduction

2nd Page, Line 20, ….. are widely used in GP,… ??? (in line 27 same)

2nd Page, Line 24, …………based on the main effects only… (in line 26 same)

2nd Page, Line 32,……… challengeto identify…

Material Methods

# The application for the crossover study to obtain hybrids can be explained. Here, only the numbers of individuals are given, there is no information on how hybridization is done.

# Why was the SNP chosen? Which primers were used? How was the protocol implemented?

3rd Page, Line 19 and 21, ….plots measuring 6.4 m2…., …..4.8-m2…. ???

3rd Page, Line 39, ….. the naïve method. After imputa….

Results

# The data obtained, especially tables and graphs, is too much, which makes it difficult to understand the article, they can be given in a more understandable way.

Discussion

# With so much data, a discussion section has been too short. The obtained data should be discussed further with similar applications in the literature.

Page 17, Line 30, ……. Jarquin et al. {15} and Xu et al. {14} used the Pearson´s correlation.

Conclusions

OK

Tables

# The tables provided in addition are very long and repetitive from time to time. Common parts can be combined into a less complex form, and they can be reorganized.

Figures

# Graphic images are not clear. Graphics should be reorganized in accordance with the page structure.

References

This section should be reorganized according to the writing rules of the journal.

Author Response

RESPONSE TO REVISION ARTICLE GENES_2132017

RESPONSE to REVIEWER 2

Dear Authors,

The manuscript entitled "Integrating parental phenotypic data enhances prediction accuracy of hybrids in wheat traits" was revised. In general, the study can be useful for researchers working in the related field; however, in the material and methods section, only the results of the methods applied for "phenotypic" and "genotypic" data are given. The mention of these methods is necessary, or at least references to these applications should be cited. For example, SNP is mentioned, but only results are in the text. Why was the SNP chosen? Which primers were used? How was the protocol implemented? A publication on genes should answer these questions. Also, for some readers of the "genes" journal, this section is explained in a very complex way, and it should be given in a more understandable way. The study seems to address the field of "data mining," where statistical data are evaluated rather than genetic approaches. It would be more appropriate to publish the manuscript after the rearrangement mentioned in the report.

 Kind regards

This revisions described below;

RESPONSE: Many thanks for investing your time reading, correcting, and suggesting changes. This is very much appreciated by all the authors.

 Title

OK

Abstract

Line 5-6, …..and additive model of (Pheno_male+Pheno_female)/2…

Line 10, it's been repeated here …..

….at least 14.1%, 5.5%, 51.4%, and 6.4% of (MD vs MD_C)…. seems more appropriate…

RESPONSE: Done.  when parental information of the same trait was used and when when parental information of the same trait and other correlated traits were used. See lines 30-31.

# Similarly, this arrangement can be made in the text.

Keywords

OK

Introduction

2nd Page, Line 20, ….. are widely used in GP,… ??? (in line 27 same)

RESPONSE: Done. See line 63.

2nd Page, Line 24, …………based on the main effects only… (in line 26 same)

RESPONSE: Done See line 74.

2nd Page, Line 32,……… challenge—to identify…

RESPONSE: Done See line 80.

Material Methods

# The application for the crossover study to obtain hybrids can be explained. Here, only the numbers of individuals are given, there is no information on how hybridization is done.

RESPONSE: Sorry we not have access to this information.

# Why was the SNP chosen? Which primers were used? How was the protocol implemented?

RESPONSE: Sorry we do not have access to this information. We just recieved the Genomic relationship matrix from the breeders.

3rd Page, Line 19 and 21, ….plots measuring 6.4 m2…., …..4.8-m2…. ???

RESPONSE: Correction done in the new version of the paper. See lines 111 and 113.

3rd Page, Line 39, ….. the naïve method. After imputa….

RESPONSE: Done. See lines 128-129.

Results

# The data obtained, especially tables and graphs, is too much, which makes it difficult to understand the article, they can be given in a more understandable way.

RESPONSE: Thanks for your suggestions. Yes, indeed we have followed up your advice and have re-written the results section in the revised version of the article.  We have moved the tables to the APPENDIX. We maintained the Figures but reduced their description to less than half their initial form.

Discussion

# With so much data, a discussion section has been too short. The obtained data should be discussed further with similar applications in the literature.

Page 17, Line 30, ……. Jarquin et al. {15} and Xu et al. {14} used the Pearson´s correlation.

RESPONSE: Thanks for your revision. Correction done in the new version of the paper. See lines 466, and 473.

Conclusions

OK

Tables

# The tables provided in addition are very long and repetitive from time to time. Common parts can be combined into a less complex form, and they can be reorganized.

RESPONSE: Thanks for your comment. As already mentioned We maintained the tables and some details were corrected but all of them form part of the APPENDIX. We have reduced to less than half the interpretation to facilitate a better understanding.

Figures

# Graphic images are not clear. Graphics should be reorganized in accordance with the page structure.

RESPONSE: We improved some details in materials and methods to clarify the figures, but we believe that the figures are really important and for this reason were maintained.

References

This section should be reorganized according to the writing rules of the journal.

RESPONSE: Yes, and thanks. We have made the corrections. See lines 565-631.

Round 2

Reviewer 2 Report

Dear Authors,

In the revised manuscript, some corrections suggested in the first report are still needed. For example, there are excessive repetitions in the table in the appendix. It would be more appropriate to correct it as I suggested in the first report. Again, it would be useful to make corrections that were not made in my first report.

Kind regards

Author Response

RESPONSE TO REVISION ARTICLE GENES_2132017

RESPONSE to REVIEWER 2

Dear Authors,

The manuscript entitled "Integrating parental phenotypic data enhances prediction accuracy of hybrids in wheat traits" was revised. In general, the study can be useful for researchers working in the related field; however, in the material and methods section, only the results of the methods applied for "phenotypic" and "genotypic" data are given. The mention of these methods is necessary, or at least references to these applications should be cited. For example, SNP is mentioned, but only results are in the text. Why was the SNP chosen? Which primers were used? How was the protocol implemented? A publication on genes should answer these questions. Also, for some readers of the "genes" journal, this section is explained in a very complex way, and it should be given in a more understandable way. The study seems to address the field of "data mining," where statistical data are evaluated rather than genetic approaches. It would be more appropriate to publish the manuscript after the rearrangement mentioned in the report.

 Kind regards

RESPONSE: we have clarified several matters that are of the reviewers’ concern. However, we cannot give details of the protocols used for applying the SNP – because WE DO NOT HAVE IT. This data emerged from a project between CIMMYT and Syngenta and the SNPs used were performed at the Syngenta Lab and the information given is the information used.

This data was published in THE PLANT GENOME journal in 2019 under Basnet et at. (2019)

Basnet, B.R.; Crossa, J.; Dreisigacker, S.; Pérez-Rodríguez, P.; Manes, Y.; Singh, R.P.; Rosyara, U.R.; Camarillo-Castillo, F.; Murua, M. Hybrid Wheat Prediction Using Genomic, Pedigree, and Environmental Covariables Interaction Models. Plant Genome. 2019, Mar;12(1). doi: 10.3835/plantgenome2018.07.0051. PMID: 30951082.

PLEASE IF THIS DOES NOT ALLOW ‘GENES’ TO PUBLISH THIS ARTICLE LET US KNOW AND WE WILL WITHRAW THE ARTICLE. WE DO NOT HAVE ANY FURTHER INFORMATION TO SHARE. CIMMYT SHARES ABSOLUTELY ALL DATA AVAILABLE AS A PUBLIC INSTITUTION. However, in most of the genomic journals this is the information given on the Materials and Methods and we had absolutely no problem with that.

We have published several articles based on Genomic Prediction in GENES. Thus, we are confused with the new REVIEWER 2 assessment.

RESPONSE: You can see on the lines highlithed in GREEN that we have changed text according with your concerns:

  1. We clarified the objective of the research at the ABSTRACT

Abstract

Genomic selection (GS) is a methodology revolutionizing plant breeding, as it can select candidates’ genotypes without phenotypic evaluation in the field. However, its practical implementation in hybrid prediction remains challenging since many factors affect its accuracy. The main objective of this research was to study the genomic prediction accuracy of wheat hybrids by adding to the model covariates with the hybrid parental phenotypic information. Four types of different models (MA, MB, MC, and MD) with one covariate (same trait to be predicted) (MA_C, MB_C, MC_C, and MD_C) or several covariates (of the same trait and other correlated traits) (MA_AC, MB_AC, MC_AC, and MD_AC) were studies. We…..

  1. We clearly explained the 4 model types (A, B, C, and D) and the manned we incorporated the parental phenotypic information. See the end of the INTRODUCTION

These 4 types of models are MA, MB, MC, and MD. We further proposed two ways of adding the parental phenotypic information as covariates to the genomic prediction models: (a) using only the parental information of the trait to be predicted (MA_C, MB_C, MC_C, and MD_C) and (b) using both parental information of the trait to be predicted and other correlated traits (MA_AC, MB_AC, MC_AC, and MD_AC).

  1. We have clarified the matter of the absolute value.

where  denotes additive,is the phenotypic value of the parental male line for the tth trait and is the phenotypic value of the parental female line for the tth trait where the male and female are assumed to belong to different heterotic groups. The other covariates capture the dominance part, and it is computed as the absolute value of () for the tth trait :

  1. We have extended the INTRODUCTION following reviewer’s advise for clarification

Prediction of hybrid performance is of fundamental importance in modern hybrid breeding programs and the best linear unbiased prediction (BLUP) model has been found useful for predicting the performance of unobserved single crosses using the performance of observed single crosses based on the pedigree (i.e., coancestry coefficient) relationship between the inbred lines forming the unobserved and observed single crosses. When studying and assessing hybrid performance two sources of variation are of importance: estimation of the additive effects among lines based on the variance of the general combining ability and the dominance and/or epistatic effects among the lines based on the variance of the specific combining ability of the cross between lines.

  1. As requested, we have moved all the 5 extensive tables to the SUPPLEMENTARY TABLES

Five Supplementary Tables are given in Tables S1-S5 to complement the results displayed in the five figures (Figures 1-5). Supplementary Tables S1-S4 had the predictions for each trait in each environment and across environments for each trait (global) for type A, B, C, and D models in terms o mean squared error under untested lines in tested environments cross validation strategies, respectively. Table S5 shows the prediction performance across environments for each trait (global) for all models MA, MA_C, MA_AC, MB, MB_C, MB_AC, MC, MC_C, MC_AC, MD, MD_C and MD_AC in terms of mean squared error (MSE) under untested lines in tested environments across validation strategy.

  1. We have extended the DISCUSSION

Hybrid breeding is an efficient system to break the yield barriers in many crops. CIMMYT elite lines derived from the spring bread wheat line breeding program are crossed and the resulting F1 hybrids are primarily evaluated in Mexico. However, as CIMMYT lacks distinct male and female breeding pools, the main challenge remains in parental selection for hybrid production. Although the per se performance of inbreds, plant phenology and cross pollinating traits, are the initial criteria to select the parents for hybrid production and testing, there is limitation on the number of combination that we actually can produce and test in the field. Since the beginning of hybrid wheat program, CIMMYT has been routinely using coefficient of parentage (COP) as one of the parental selection criteria in order to maximize genetic diversity. In the hybrid study of Basnet [16] it is demonstrated the potential application of pedigree information and molecular marker data in predicting single cross hybrids performance. Applying HP, we envision selecting potential hybrid parents from a broader genetic germplasm pool and earlier in the breeding cycle after preliminary and elite yield trail evaluation out of CIMMYT elite breeding lines. However, the hybrid prediction study of Basnet [16] did not inlcude any covariate using parental information to enhance hybrid prediction. This study employing wheat hybrid data from data from Basnet [16] show the increase in genomic prediction accuracy by using existing parental phenotypic data. Further studies are requiered to include enviromental studies together with parental phenotypic information.

Furthermore, results of this study show that adding the parental phenotipic information to the genomic prediction model enhances the prediction accuracy of the GS methodology. It is noted that the way the parental phenotypic information is incorporated in the genomic prediction models is key to improve the prediction performace. Also we found that only a small gain in prediction performance is reached when, in addition to the parental phenotypic information of the same trait to be predicted, it is also added the parental phenotypic information of other correlated traits. This results could be interpreted that adding other correlated traits from parents as covariates can be helpful to increase the prediction accuracy only in those cases when these other traits are highly correlated with the trait of interest (to be predicted); otherwise if the degree of correlation is low no further imporvement on genomic prediction accuracy are achieved.

  1. We have added the related information on the Supplementary Tables

Supplementary Tables: Five supplementary tables can be downloaded at: www.mdpi.com/xxx/s1 with results Table S1. Prediction performance for each trait in each environment and across environments for each trait (global) for type A models (MA, MA_C and MA_AC) in terms of mean squared error (MSE) under untested lines in tested environments across validation strategy. The Relative efficiency (RE_MSE) were computed dividing the MSE of model MA by the MSE of model MA_C, also was computed the RE_MSE for comparing models MA vs MA_AC and for comparing models MA_C vs MA_AC. Table S2. Prediction performance for each trait in each environment and across environments for each trait (global) for type B models (MB, MB_C and MB_AC) in terms of mean squared error (MSE) under untested lines in tested environments across validation strategy. The Relative efficiency (RE_MSE) were computed dividing the MSE of model MB by the MSE of model MB_C, also was computed the RE_MSE for comparing models MB vs MB_AC and for comparing models MB_C vs MB_AC. Table S3. Prediction performance for each trait in each environment and across environments for each trait (global) for type C models (MC, MC_C and MC_AC) in terms of mean squared error (MSE) under untested lines in tested environments across validation strategy. The Relative efficiency (RE_MSE) were computed dividing the MSE of model MC by the MSE of model MC_C, also was computed the RE_MSE for comparing models MC vs MC_AC and for comparing models MC_C vs MC_AC. Table S4. Prediction performance for each trait in each environment and across environments for each trait (global) for type D models (MD, MD_C and MD_AC) in terms of mean squared error (MSE) under untested lines in tested environments across validation strategy. The Relative efficiency (RE_MSE) were computed dividing the MSE of model MD by the MSE of model MD_C, also was computed the RE_MSE for comparing models MD vs MD_AC and for comparing models MD_C vs MD_AC. Table S5. Prediction performance across environments for each trait (global) for all models MA, MA_C, MA_AC, MB, MB_C, MB_AC, MC, MC_C, MC_AC, MD, MD_C and MD_AC in terms of mean squared error (MSE) under untested lines in tested environments across validation strategy. The Relative efficiency (RE_MSE) were computed dividing the MSE of model MX by the MSE of model MX_C or MX_AC model, with X taking the values of A, B, C and D.

This revisions described below;

RESPONSE: Many thanks for investing your time reading, correcting, and suggesting changes. This is very much appreciated by all the authors.

 Title

OK

Abstract

Line 5-6, …..and additive model of (Pheno_male+Pheno_female)/2…

Line 10, it's been repeated here …..

….at least 14.1%, 5.5%, 51.4%, and 6.4% of (MD vs MD_C)…. seems more appropriate…

RESPONSE: Done.  when parental information of the same trait was used and when when parental information of the same trait and other correlated traits were used. See lines 30-31.

# Similarly, this arrangement can be made in the text.

Keywords

OK

Introduction

2nd Page, Line 20, ….. are widely used in GP,… ??? (in line 27 same)

RESPONSE: Done. See line 63.

2nd Page, Line 24, …………based on the main effects only… (in line 26 same)

RESPONSE: Done See line 74.

2nd Page, Line 32,……… challenge—to identify…

RESPONSE: Done See line 80.

Material Methods

# The application for the crossover study to obtain hybrids can be explained. Here, only the numbers of individuals are given, there is no information on how hybridization is done.

RESPONSE: Sorry we not have access to this information.

# Why was the SNP chosen? Which primers were used? How was the protocol implemented?

RESPONSE: Sorry we do not have access to this information. We just recieved the Genomic relationship matrix from the breeders.

3rd Page, Line 19 and 21, ….plots measuring 6.4 m2…., …..4.8-m2…. ???

RESPONSE: Correction done in the new version of the paper. See lines 111 and 113.

3rd Page, Line 39, ….. the naïve method. After imputa….

RESPONSE: Done. See lines 128-129.

Results

# The data obtained, especially tables and graphs, is too much, which makes it difficult to understand the article, they can be given in a more understandable way.

RESPONSE: Thanks for your suggestions. Yes, indeed we have followed up your advice and have re-written the results section in the revised version of the article.  We have moved the tables to the APPENDIX. We maintained the Figures but reduced their description to less than half their initial form.

Discussion

# With so much data, a discussion section has been too short. The obtained data should be discussed further with similar applications in the literature.

Page 17, Line 30, ……. Jarquin et al. {15} and Xu et al. {14} used the Pearson´s correlation.

RESPONSE: Thanks for your revision. Correction done in the new version of the paper. See lines 466, and 473.

Conclusions

OK

Tables

# The tables provided in addition are very long and repetitive from time to time. Common parts can be combined into a less complex form, and they can be reorganized.

RESPONSE: Thanks for your comment. As already mentioned We maintained the tables and some details were corrected but all of them form part of the APPENDIX. We have reduced to less than half the interpretation to facilitate a better understanding.

Figures

# Graphic images are not clear. Graphics should be reorganized in accordance with the page structure.

RESPONSE: We improved some details in materials and methods to clarify the figures, but we believe that the figures are really important and for this reason were maintained.

References

This section should be reorganized according to the writing rules of the journal.

RESPONSE: Yes, and thanks. We have made the corrections. See lines 565-631.
